# Oxidative Stress in HIV-Associated Neurodegeneration: Mechanisms of Pathogenesis and Therapeutic Targets

**DOI:** 10.3390/ijms26146724

**Published:** 2025-07-13

**Authors:** Sophia Gagliardi, Tristan Hotchkin, Grace Hillmer, Maeve Engelbride, Alexander Diggs, Hasset Tibebe, Coco Izumi, Cailyn Sullivan, Cecelia Cropp, Olive Lantz, Dacia Marquez, Jason Chang, Jiro Ezaki, Alexander George Zestos, Anthony L. Riley, Taisuke Izumi

**Affiliations:** 1Department of Biology, College of Arts & Sciences, American University, Washington, DC 20016, USA; sg1978a@american.edu (S.G.); th9716a@american.edu (T.H.); gh2297a@american.edu (G.H.); me6273a@american.edu (M.E.); ad6045a@american.edu (A.D.); ht8146a@american.edu (H.T.); cizumi@american.edu (C.I.); cs0973a@american.edu (C.S.); cc4343a@american.edu (C.C.); ol3810a@american.edu (O.L.); dm6732a@american.edu (D.M.); jchang@american.edu (J.C.); jezaki@american.edu (J.E.); 2Department of Chemistry, College of Arts & Sciences, American University, Washington, DC 20016, USA; zestos@american.edu; 3Department of Neuroscience, College of Arts & Sciences, American University, Washington, DC 20016, USA; alriley@american.edu; 4District of Columbia Center for AIDS Research, Washington, DC 20052, USA

**Keywords:** human immunodeficiency virus type 1 (HIV-1), reactive oxygen species (ROS), HIV-associated neurocognitive disorders (HAND), central nervous system (CNS), Alzheimer’s Disease (AD)

## Abstract

Treatment for HIV infection has become more manageable due to advances in combination antiretroviral therapy (cART). However, HIV still significantly affects the central nervous system (CNS) in infected individuals, even with effective plasma viral suppression, due to persistent viral reservoirs and chronic neuroinflammation. This ongoing inflammation contributes to the development of HIV-associated neurocognitive disorders (HANDs), including dementia and Alzheimer’s disease-like pathology. These complications are particularly prevalent among the aging population with HIV. This review aims to provide a comprehensive overview of HAND, with a focus on the contribution of oxidative stress induced by HIV-mediated reactive oxygen species (ROS) production through viral proteins such as gp120, Tat, Nef, Vpr, and reverse transcriptase. In addition, we discuss current and emerging therapeutic interventions targeting HAND, including antioxidant strategies and poly (ADP-ribose) polymerase (PARP) inhibitors. These are potential adjunctive approaches to mitigate neuroinflammation and oxidative damage in the CNS.

## 1. Introduction

Human Immunodeficiency Virus type 1 (HIV-1) is still one of the most significant global public health challenges [1]. As of 2023, the World Health Organization (WHO) reported that there are 39.9 million people living with HIV (PLWH) worldwide, with approximately 630,000 deaths attributed to HIV-related illnesses [2,3]. If left untreated, HIV can progress to acquired immunodeficiency syndrome (AIDS), resulting in death due to the inability of the immune system to fight infection [4,5]. Combination antiretroviral therapy (cART), which employs multiple antiviral agents to inhibit various stages of the HIV replication cycle, has revolutionized the treatment of HIV [6]. cART effectively suppresses plasma viral load to undetectable levels (<50 copies/mL), significantly improving long-term survival and overall health outcomes in individuals infected with HIV. Additionally, the development of long-acting antivirals, such as cabotegravir and lenacapavir, has further improved the quality of life for PLWH by reducing dosing frequency and enhancing adherence [7,8]. Despite these advancements in controlling viremia, a complete cure for HIV remains elusive, as the virus persists in latent reservoirs within host cells despite cART. Notably, even in individuals with well-controlled plasma viral loads on cART, HIV can significantly affect the central nervous system (CNS), leading to HIV-associated neurocognitive disorders (HANDs), a spectrum of cognitive impairments ranging from asymptomatic or mild deficits to severe HIV-associated dementia (HAD) [9,10]. While the incidence of HAD has declined because of widespread cART use, milder forms of cognitive impairment remain prevalent, affecting up to 50% of treated individuals [11,12]. As the HIV-positive population continues to age, there is increasing concern about the intersection of chronic HIV infection with age-related neurodegenerative diseases, including Alzheimer’s disease (AD), AD-related dementias (ADRD), Lewy body disease (LBD), vascular contributions to cognitive impairment and dementia (VCID), and frontotemporal dementia (FTD) [13]. Neurodegenerative diseases such as AD and FTD are prevalent in the general population, independent of HIV infection. These conditions arise from a complex interplay of genetic predispositions, environmental exposures, and lifestyle factors, many of which contribute to chronic inflammation within the CNS [14,15,16,17,18,19,20]. A key driver of this inflammation is the activation of microglia cells, myeloid-derived immune cells that act as the primary resident macrophages of the CNS [21,22,23,24]. Microglia cells play essential roles in immune surveillance, regulation of inflammation, and the maintenance of neural homeostasis. When activated, they release proinflammatory cytokines, such as interleukin-1β (IL-1β) and tumor necrosis factor-alpha (TNF-α), which induce the neuroinflammatory milieu [15,22]. Neuroinflammation in the brain leads to the release of harmful molecules, including reactive oxygen species (ROS), which can damage neurons and impair synaptic communication, ultimately disrupting overall brain function. This damage can contribute to the development of neurodegenerative diseases [22,25,26]. As aging occurs, these microglial cells tend to be activated more frequently, thus resulting in an increased release of proinflammatory cytokines that damage the brain. In aging populations, one factor contributing to microglial activation is the accumulation of misfolded proteins, such as amyloid-β (Aβ) plaques and neurofibrillary tangles, hallmark features of neurodegenerative diseases that appear early in the course of AD [26,27,28].

Microglial cells are primary targets of HIV infection and serve as the predominant viral reservoir within brain tissue [29,30,31,32,33]. Even in individuals with suppressed plasma viral loads on cART, HIV can promote a chronic inflammatory environment in the CNS by activating microglial cells, contributing to neuronal dysfunction and degeneration [34,35,36]. This microglia-mediated neuroinflammation is considered a primary driver of HAND pathogenesis [34,35,36,37]. A primary mechanism underlying HIV-induced neuroinflammation is the production of ROS by microglia triggered by viral infection or exposure to secreted viral proteins. In this review, we explore the molecular mechanisms of ROS generation and their role in driving neuroinflammation and neurodegeneration in the context of HAND, while also discussing emerging therapeutic strategies aimed at improving health outcomes for individuals living with chronic HIV infection.

## 2. Neurocognitive Disorders Caused by HIV Infection

HIV-related neurodegeneration is mediated via multiple mechanisms. The first of these mechanisms involves the virus and subsequent viral proteins that are secreted into the plasma and infiltrate the CNS by crossing the blood–brain barrier (BBB) (Figure 1[I]) [38,39]. This often occurs via infected monocytes that act as ‘Trojan horses’, which can establish a persistent viral reservoir in the brain (Figure 1[I]) [34]. HIV-infected monocytes and secreted viral proteins, such as gp120 and Tat, have been shown to disrupt BBB integrity and increase its permeability (Figure 1[I]) [34,40]. As a result of a compromised BBB, immune cells, viral particles, and neurotoxic HIV proteins can further infiltrate the CNS, leading to sustained inflammation and oxidative damage (Figure 1[II]) [38,41]. This breakdown of the barrier contributes to a self-reinforcing cycle of immune cell infiltration, sustained neuroinflammation, and progressive neurodegenerative changes. Even with effective antiretroviral therapy, a substantial proportion of individuals with HIV exhibit ongoing BBB dysfunction [42,43,44].

HIV infection of the brain leads to ongoing activation of resident immune cells, such as microglia and infiltrating macrophages [32,45]. Consistently, elevated levels of cytokines and chemokines can progressively damage neurons and synapses, leading to neuronal and synaptic dysfunction. As a result of this chronic immune activation, the production of ROS by infected immune cells is increased, establishing a self-perpetuating cycle of inflammation and oxidative stress [46,47]. This neuroinflammation is a central feature of HAND and can synergize with other neurodegenerative processes, such as mitochondrial dysfunction and the accumulation of misfolded proteins like Aβ and hyperphosphorylated tau (Figure 2) [48,49,50,51,52,53,54,55]. Oxidative stress has emerged as a pivotal mechanism in HIV-related neurocognitive impairment. HIV infection and its inducible inflammation disrupt the balance of reduction and oxidation reactions within the cell, leading to excessive ROS production in the CNS [56,57]. Viral proteins, particularly gp120, Tat, Nef, and Vpr, also initiate a cascade of harmful processes by exerting direct neurotoxic effects on neurons and glial cells [46]. Additionally, these proteins stimulate the production of ROS and proinflammatory cytokines within the brain [56]. The detailed mechanisms of viral protein-mediated ROS production are discussed in Section 3.

## 3. Mechanisms of Reactive Oxygen Species (ROS) Production by Human Immunodeficiency Virus

### 3.1. HIV Pathogenesis and ROS Production

HIV-1 is specifically known to induce the production of ROS in host cells, contributing to the pathogenesis of the virus [47,58]. As a result, HIV infection is associated with elevated oxidative stress in host cells. In both infected cells and bystander cells, HIV-1 triggers the overproduction of ROS compared to normal, healthy conditions [47,59]. Multiple HIV proteins, including the envelope glycoprotein (gp120), the regulatory protein (Tat), some accessory proteins (Nef and Vpr), and even the viral enzyme, RT, have been shown to stimulate ROS generation in host cells (Figure 3A) [58,60,61,62,63,64]. In addition to the viral proteins, there are other factors of HIV infection that can lead to the increased production of ROS. HIV infection involves the production of viral cDNA, including linear and circular forms such as 2-LTR circles [65]. HIV-derived cDNAs, particularly 2-LTR unintegrated forms, can activate the cGAS-STING pathway, which is known to drive innate immune responses and chronic inflammation [66]. This activation leads to increased production of ROS and oxidative stress, which can trigger ferroptosis, a distinct, iron-dependent form of regulated cell death characterized by the accumulation of lipid peroxides and membrane damage [67,68,69]. This pathway may also contribute to neurotoxicity in the context of HAND. Furthermore, prolonged use of cART, particularly regimens containing nucleoside RT inhibitors, has been associated with increased oxidative stress, which may contribute to neuronal injury and cognitive dysfunction [70,71]. This virus infection-induced oxidative stress plays a significant role in HIV pathogenesis, contributing to both direct and indirect effects, such as cellular damage and chronic inflammation. Notably, a persistent imbalance in the host redox state has been recognized as a key factor in various HIV-associated conditions, such as neurocognitive disorders and immune dysfunction, underlining the importance of a more comprehensive understanding of the role of ROS in HIV disease progression [47]. In the following subparagraph, we will discuss in more detail how each viral protein contributes to ROS production.

### 3.2. Envelope Glycoprotein (gp120) and Tat Mediated ROS Production

Several studies have shown that exposure to the HIV-1 viral envelope glycoprotein gp120 leads to increased ROS production and triggers intracellular oxidative stress in neural and immune cells, even in the absence of full viral infection. The glycoprotein gp120 stimulates oxidative stress in microglia and astrocytes, leading to the release of proinflammatory cytokines, such as IL-1β (Figure 1[II]), that disrupt neuronal calcium homeostasis and cause neuron death (Figure 1[III]) [64,72,73]. Specifically, Viviani et al. demonstrated that exposing glial cells to HIV gp120 leads to a marked increase in ROS production within those cells [60]. Additionally, their subsequent research indicated that gp120 induces IL-1β release from glial cells, which further contributes to neurotoxicity, highlighting a cellular-level mechanism for gp120-mediated production of ROS (Figure 3B) [74]. IL-1β, released by gp120-activated glial cells, subsequently disrupts calcium homeostasis in nearby neurons by enhancing N-methyl-D-aspartate receptor (NMDAR) activity, ultimately leading to neuronal injury and death [58,75]. Importantly, antioxidant-mediated neutralization of ROS prevented neuronal damage in a sandwich co-culture of primary hippocampal neurons and glial cells, confirming that the neurotoxicity was driven by ROS released from gp120-activated glia [60]. This study revealed an oxidative cascade initiated by gp120 in glial cells, characterized by ROS generation and IL-1β release, that subsequently disrupts the calcium homeostasis while also inducing neuronal death in adjacent cells (Figure 3B) [60]. Such a mechanism implicates oxidative stress as a driver of HIV-related neurotoxicity.

Further investigations found that certain neuron populations are especially vulnerable to gp120-induced oxidative stress. Agrawal et al. reported that dopamine-producing neurons (dopaminergic neurons), such as those in the substantia nigra of the brain, are particularly sensitive to oxidative stress caused by gp120 [76]. Dopaminergic neurons cultured in vitro underwent apoptosis at much lower peroxide concentrations than other neurons, indicating a heightened sensitivity to ROS. Correspondingly, when these neurons were exposed to HIV-1 gp120, they showed significant oxidative damage and apoptosis. This effect was mediated by gp120-induced calcium influx in neurons, with a selective vulnerability observed in dopaminergic cells, indicating that these neurons were more susceptible to gp120-induced toxicity than other neuronal subtypes. Dopaminergic neurons play a central role in the neurocognitive impairments associated with chronic HIV infection, as evidenced by reduced peak dopamine levels and slower reuptake rates that are linked to motivational and cognitive deficits such as apathy and depression, which are core features of HANDs [77]. Crucially, antioxidant interventions protect the cells [73,75,78]. The pretreatment with the antioxidant N-acetylcysteine (NAC) or the overexpression of antioxidant enzymes, such as superoxide dismutase (SOD) and glutathione peroxidase, blocked the gp120-induced cell death [76]. In vivo experiments reinforced these findings. When low doses of gp120 were injected into the brains of rats using a stereotaxic frame, it caused a rapid loss of dopaminergic neurons—an effect that was prevented by antioxidant enzyme treatment [79]. These findings provide compelling evidence that in vitro exposure to gp120 in primary human astrocytes induces ROS and contributes to selective neurotoxicity. While the role of gp120 in promoting ROS production and neurotoxicity is well established, more recent studies have elucidated the upstream molecular pathways involved. Ivanov et al. demonstrated that gp120 enhances ROS generation in astrocytes through multiple parallel mechanisms [47]. These include the upregulation of cytochrome P450 2E1 (CYP2E1), activation of NADPH oxidases NOX2 and NOX4, and promotion of Fenton and Weiss-Haber reactions—all of which contribute to oxidative stress [47,80,81,82]. Additionally, Pandhare et al. provided mechanistic evidence linking gp120 exposure to ROS generation in neuroblastoma cells [83]. They demonstrated that gp120 activates the tumor suppressor protein p53 via the CXCR4 co-receptor, which subsequently upregulates proline oxidase (POX). This cascade leads to increased ROS production and autophagic activity. Inhibition of POX reduced both ROS levels and autophagy, whereas POX overexpression amplified these responses.

The Tat protein can persist in the CNS despite cART and has been shown to promote AD-related pathology by binding to Aβ, disrupting its metabolism, and inducing tau hyperphosphorylation (Figure 2). These effects accelerate the formation of amyloid plaques and neurofibrillary tangles [84,85]. Such neurotoxic viral proteins contribute to synaptic damage, neuronal loss, and oxidative stress, driving the development of HAND and potentially increasing the risk of other dementias in individuals chronically living with HIV.

### 3.3. Nef-Mediated ROS Production

In addition to gp120 and Tat, the HIV accessory proteins Nef and Vpr significantly contribute to ROS induction within host cells, which further amplifies HIV-associated pathogenesis [86,87,88]. It has been established that Nef promotes oxidative stress by dysregulating the host cell’s NADPH oxidase complexes and mitochondrial function. One of the primary mechanisms is the interaction of Nef with components of the NADPH oxidase complex in macrophages and microglia [86,89]. Nef has been found to be associated with p22-phox, a subunit of the NADPH oxidase complex, leading to enhanced production of superoxide anions (Figure 3C[I]) [90]. This interaction effectively primes oxidase. In addition, Nef triggers intracellular kinases, including Src family kinases and PI3K, that lead to phosphorylation of p47-phox, which is a key cytosolic subunit of NADPH oxidase in the U937 human monoblastic cell line [91].

While expression of Nef did not activate the NADPH oxidase by itself in the microglia cell model, it is able to indirectly activate the NADPH oxidase by triggering the Vav/Rac1/p21-activated kinase (PAK) signaling pathway (Figure 3C[I]) [89]. This pathway is critical, as it plays a key role in initiating NADPH oxidase activity. This signaling cascade primes the cells for an amplified oxidative response upon subsequent stimulation, such as calcium ionophores, formyl peptide, or endotoxins [89]. Nef interacts with the guanine nucleotide exchange factor Vav, which in turn facilitates the exchange of GDP for GTP on Rac1, converting it to its active GTP-bound form. This activation of Rac1 leads to the stimulation of PAK signaling, culminating in the priming of the NADPH oxidase complex (gp91) for enhanced ROS production (Figure 3C[I]) [89,92]. Mutations in Nef that disrupt its interaction with this pathway abolish its ability to enhance ROS production, confirming that Nef-mediated activation of the PAK pathway is responsible for the increased superoxide release. Interestingly, Nef modulates ROS production in a biphasic manner, initiating an early increase in superoxide levels that is subsequently followed by a decline in NADPH oxidase activity [86]. This reduction has been attributed to Nef-induced soluble factors, such as IL-10, which suppress oxidase function. Overall, this highlights Nef’s complex regulatory influence on the cellular redox balance (Figure 3C[II]) [86]. The oxidative environment induced by Nef contributes to chronic inflammation by promoting inflammasome activation, which triggers pyroptosis in uninfected bystander CD4 T cells. This mechanism underlies the characteristic T-cell depletion observed in HIV infection (Figure 3C[III]), while also leading to the secretion of proinflammatory cytokines, such as IL-1β and IL-18, which sustain the aforementioned inflammatory milieu (Figure 3C[IV]) [93].

### 3.4. Vpr-Mediated ROS Production

Similarly, another accessory protein, Vpr, exerts potent oxidative effects by targeting mitochondria (Figure 3D) [88]. It localizes to mitochondrial membranes and interacts with components of the permeability transition pore complex, including the adenine nucleotide translocator (ANT) in the inner membrane and the voltage-dependent anion channel (VDAC) in the outer membrane. Vpr enters the mitochondria via the VDAC and binds to ANT, triggering the depolarization of the mitochondrial membrane. This results in an increase in the leakage of reactive species from the mitochondria (Figure 3D) [94]. Accordingly, Vpr can induce mitochondrial membrane permeabilization and uncouple the respiratory chain, leading to electron leakage and the subsequent elevation of ROS production. Vpr-induced mitochondrial membrane permeabilization also contributes to the release of pro-apoptotic factors, such as cytochrome c and apoptosis-inducing factor (AIF) from mitochondria into the cytosol, which activates the caspase cascade and apoptosis [88,95,96,97,98]. The brain and kidneys are particularly vulnerable to Vpr-induced toxicity, as Vpr has been implicated in HIV-associated neurodegeneration and nephropathy, largely through oxidative stress in neurons and renal tubular cells [99,100,101,102]. Importantly, Vpr-induced oxidative stress not only causes cell death but also can paradoxically stimulate HIV-1 replication. The excess ROS generated by Vpr activates redox-sensitive transcription factors, such as NF-κB and AP-1, which then enhance HIV gene expression (Figure 3D) [103,104]. Sandoval et al. recently demonstrated that Vpr-inducible DNA damage activates NF-κB through the DNA damage sensor ATM and the signaling adaptor NEMO, independently of the effects of Vpr on the cell cycle [105]. This NF-κB activation leads to increased HIV-1 transcription and promotes the production of proinflammatory cytokines, contributing to chronic immune activation in HIV infection [106]. Conversely, an antioxidant-rich environment can suppress HIV transcription. Israël et al. investigated whether shifting the cellular redox balance toward a more reduced state using butylated hydroxyanisole (BHA) could downregulate HIV promoter activity in lymphoblastoid and monocytic cell lines [107]. Their findings showed that antioxidant or free radical scavenger treatment in HIV-infected T cells and monocytes inhibited NF-κB nuclear translocation and reduced its activation, thereby suppressing HIV transcription. This led to a significant decrease in HIV long terminal repeat (LTR) promoter activity and consequently lower viral production. The drop in viral replication was accompanied by reduced secretion of NF-κB–dependent proinflammatory cytokines. These findings reveal a feed-forward loop in which HIV-induced ROS boosts NF-κB, which in turn enhances HIV replication and inflammatory gene expression.

### 3.5. Reverse Transcriptase (RT) Mediated ROS Production

While the HIV proteins, gp120, Tat, Nef, and Vpr, have well-documented roles in promoting oxidative stress, recent studies have also highlighted the role of HIV-1 RT in ROS generation [108,109]. RT is the viral enzyme responsible for converting RNA to DNA, but when expressed in host cells, it can have additional unexpected effects on cellular metabolism. Notably, expression of HIV-1 RT in mammalian cells has been shown to lead to significant ROS production (Figure 3E). Zakirova et al. demonstrated that introducing HIV-1 RT into a murine mammary carcinoma cell line (4T1-luc2) caused a marked increase in ROS levels, along with elevated lipid peroxidation and enhanced cell motility [62]. These effects were associated with the upregulation of epithelial–mesenchymal transition (EMT) markers, such as Twist, suggesting a link between RT-induced oxidative stress and pro-tumorigenic cellular changes (Figure 3E) [109]. Strikingly, RT variants with drug-resistance mutations that impair polymerase activity produced significantly less ROS, consequently losing their ability to promote tumor growth and metastasis [109]. In immunodeficient mice, tumors derived from cells expressing wild-type RT grew faster and formed more metastases than those expressing mutant RT, correlating with higher Twist expression and oxidative stress in wild-type RT tumors. These findings underscore that the enzymatic activity or structural integrity of HIV-1 RT is essential for driving ROS production and EMT-associated tumor progression. Additional evidence for RT mediated oxidative impact comes from studies in human cell lines. Isaguliants et al. reported that transient expression of HIV-1 RT in the human embryonic kidney cell line, HEK293 cells, triggers robust ROS generation and a cellular antioxidant response [108]. They observed a 10- to 15-fold increase in the transcription of detoxification enzymes, such as heme oxygenase-1 (HO-1), indicating significant oxidative stress in RT-expressing cells (Figure 3E). Notably, different RT gene variants produced varying levels of ROS, which in turn influenced both the magnitude and quality of the immune response elicited by a DNA vaccine targeting HIV-1 RT in their experimental model [108]. It was concluded that RT can modulate the host cellular environment through oxidative mechanisms, potentially contributing to viral pathogenesis and even oncogenic processes in cells where it is expressed.

## 4. Reactive Oxygen Species (ROS) and Chronic HIV-Associated Neurodegeneration

HIV-induced ROS production is not limited to acute neuronal injuries, but it also plays a role in the chronic neurological complications of HIV infection through downstream effects contributing to neural damage. While acute ROS production by gp120 causes immediate cellular damage, chronic oxidative stress produces long-term neurodegeneration in PLWH. In individuals with HAND, there is evidence of ongoing oxidative damage and disrupted lipid metabolism in the brain. Haughey et al. examined postmortem brain tissues and cerebrospinal fluid from individuals with HIV dementia and found clear signs of oxidative stress [110]. In these individuals, the levels of 4-hydroxynonenal (4-HNE), a reactive byproduct of lipid peroxidation, were significantly elevated in brain tissue and cerebrospinal fluid. 4-HNE has been shown to contribute to premature apoptosis via adduct formation, inhibition of survival pathways, and activation of apoptotic proteins, ultimately promoting neurodegenerative diseases such as Parkinson’s and Alzheimer’s [111,112]. Alongside 4-HNE, levels of ceramide, a sphingolipid known to promote apoptosis, were also increased [110]. Intracellular levels of ceramide increase when the activity of neutral sphingomyelinase (nSMase) or acid sphingomyelinase (aSMase) is elevated, as these enzymes convert sphingomyelin to ceramide (Figure 4) [113,114]. To explore causation, they exposed cultured neurons to the HIV-1 gp120 and Tat and observed that both provoked an accumulation of 4-HNE and ceramide within neurons (Figure 4) [110]. These findings suggest that oxidative stress and inflammatory signals triggered by HIV proteins can promote sphingomyelin breakdown and ceramide accumulation in neural cells. Given that ceramide is a potent mediator of programmed cell death, this pathway provides a mechanistic link between HIV-induced ROS and neuronal degeneration. Supporting this connection, pharmacological inhibition of ceramide synthesis protected neurons from gp120/Tat-induced toxicity [110]. These findings indicate that viral proteins induce oxidative stress in chronic HIV-associated brain disease that initiates a cascade of damaging lipid signaling, leading to excessive ceramide accumulation and subsequent apoptosis of neural cells (Figure 4). Ceramide generation mediated by nSMase has been reported to play a critical role in the late stages of HIV-1 maturation and replication, a key process required for the virus to acquire infectivity [7,115,116,117,118]. These findings suggest that ceramide generation broadly supports HIV infection at both cellular and molecular levels. This mechanism also provides a plausible explanation for how chronic HIV infection contributes to different neuropathological conditions, including dementia, through oxidative stress-driven pathways [7,115,116,117]. Additionally, the findings of Bachis et al. relating to the rapid loss of dopaminergic neurons in rat brains after low doses of gp120 were injected suggest a potential mechanism underlying the Parkinson’s disease-like symptoms occasionally observed in PLWH, wherein the gp120 protein may contribute to dopaminergic neuron loss through oxidative stress [79]. This ultimately links HIV infection to neurodegenerative processes.

There is substantial overlap between the mechanisms of HAND and those of AD. Chronic oxidative stress and inflammation in HIV can engage the same pathways that drive classic AD pathology. Epidemiological analyses further suggest that older people living with HIV may have a higher incidence of AD-like cognitive impairment compared to uninfected populations [13,56]. However, it remains under investigation whether HIV accelerates actual AD pathology or if observed AD-like symptoms in HIV are a byproduct of prolonged inflammation and vascular injury [13].

It is interesting to note how HIV itself might directly contribute to AD pathology. As mentioned above, the HIV Tat protein can bind to Aβ, impairing its clearance, and can trigger tau phosphorylation cascades (Figure 1[III]) [84]. On the other hand, some antiretroviral drugs may influence amyloid metabolism and could either mitigate or, in some cases, contribute to amyloid/tau accumulation (Figure 2) [119,120]. This complex interplay is the subject of active research, as studies attempt to discern whether long-term HIV-1 infection serves as a precipitating factor for the earlier onset of AD or other dementias. Neurocognitive issues also persist in HIV-infected individuals who are well-treated, highlighting the need for adjunctive strategies targeting these pathogenic mechanisms [121,122]. Approaches to reduce chronic inflammation and oxidative stress in PLWH are currently being explored. Although cART effectively suppresses plasma viral load over the long term, it does not fully protect the brain from the detrimental effects of chronic HIV infection [121,122]. One of the primary challenges is the limited penetration of cART into the CNS, largely due to the BBB, a tightly regulated structure composed of endothelial cells joined by tight junctions that restrict the passage of many substances, including several antiretroviral drugs [123]. This selective permeability can result in subtherapeutic drug concentrations in the brain, allowing HIV to persist and potentially develop resistance within CNS reservoirs [124,125]. In addition, some antiretroviral compounds have been associated with neurotoxic effects [126], which may contribute to neurocognitive impairments in PLWH, even when systemic viral suppression is achieved. Consequently, antioxidant therapies, such as glutathione precursors or NAC, have shown promise in preclinical models of HIV neurotoxicity through the attenuation of ROS levels and preventing neuron death caused by HIV proteins [75,78]. However, several challenges have limited their clinical efficacy, such as the same limited BBB penetration as cART, inconsistent clinical outcomes, potential pro-oxidant effects at high doses, and the complexity of HAND pathogenesis [127,128,129]. These challenges underscore the need for a comprehensive approach to treating HAND, potentially involving combination therapies that align with broader HIV cure strategies. In addition to targeting oxidative stress, anti-inflammatory treatments that reduce microglial activation have the potential to help preserve cognitive function. Ultimately, a more holistic strategy that aims to suppress viral replication, in addition to protecting long-term brain health, is essential. Details of current and future approaches for therapeutic intervention in HAND are discussed in Section 5. By addressing the overlapping mechanisms of HIV neuropathogenesis and AD, such as chronic inflammation and neuronal injury, it may be possible to reduce the risk of dementia among PLWH. Ongoing research continues to translate these mechanistic insights into therapeutic interventions, with the goal of ensuring that PLWH not only live longer but also maintain their cognitive vitality as they age.

## 5. Therapeutic Interventions for HIV-Associated Neurocognitive Disorder

### 5.1. Antioxidant

Given the detrimental effects of excess ROS production induced by various HIV-1 proteins, as discussed in Section 3 and Section 4, researchers have investigated antioxidant strategies as potential adjunctive therapies for HAND. Various preclinical studies have explored whether bolstering antioxidant defenses can mitigate HIV-induced oxidative stress (Figure 5 and Table 1). Agrawal et al. demonstrated that delivering exogenous antioxidant enzymes can protect cells from HIV protein-induced toxicity [130]. Using SV40 viral vectors, they introduced the genes for superoxide dismutase 1 (SOD1) and glutathione peroxidase 1 (GPx1) into primary human neurons, which were subsequently exposed to HIV-1 Tat. Co-expression of SOD1 and GPx1 effectively blocked Tat-induced calcium influx and neuronal apoptosis, whereas either enzyme alone provided insufficient protection. These results suggest that both superoxide and hydrogen peroxide, ROS intermediates detoxified by SOD and GPx, respectively, are involved in Tat-mediated neurotoxic signaling. Manda et al. provided clear experimental evidence supporting the protective effects of antioxidants in the context of HAND, particularly against cART-induced oxidative stress [131]. They demonstrated that cART, including azidothymidine (AZT) and indinavir (IDV), significantly induced oxidative stress and mitochondrial dysfunction in BBB endothelial cells (hCMEC/D3), leading to reduced intracellular glutathione, increased lipid peroxidation, mitochondrial membrane depolarization, ATP depletion, and apoptosis (Figure 5). Pretreatment with NACA, a cell-permeable thiol antioxidant, significantly reduced ROS levels, restored glutathione, preserved mitochondrial membrane potential, and prevented apoptosis of endothelial cells (Table 1). These findings suggest that antioxidants like NACA may protect the BBB from drug-induced oxidative stress, potentially helping to mitigate HAND in individuals treated with cART. In addition, Teodorof-Diedrich et al. provided strong experimental evidence supporting the therapeutic potential of antioxidants in mitigating neurocognitive damage associated with HIV and methamphetamine exposure [132]. In their study, NAC significantly reduced ROS levels in SK-N-MC cells, a human neuroepithelioma cell line, following exposure to HIV proteins, including gp120 and Tat, and methamphetamine. This antioxidant effect helped prevent mitochondrial fragmentation, supported autophagic processes, preserved mitochondrial integrity and function, and ultimately protected neuronal architecture. These in vitro mechanistic findings provide a strong rationale for translating antioxidant-based interventions into clinical research for HAND. Allard et al. conducted a double-blind, placebo-controlled trial to evaluate whether enhancing the antioxidant defenses of HIV-positive individuals could influence the disease progression (Figure 5 and Table 1) [133]. In this study, 49 PLWH were randomized to receive either a placebo or a daily oral supplementation of vitamin E and vitamin C, two well-known dietary antioxidants, for three months. The antioxidant-supplemented group showed a significant decrease in lipid peroxidation markers compared to controls, indicating that the vitamin regimen successfully lowered systemic oxidative stress. There was also a trend toward lower HIV viral load in the antioxidant-treated group after the supplementation period. Although this reduction in viral load was modest and did not reach statistical significance, it suggested a potentially beneficial effect. Another randomized, double-blind, placebo-controlled trial evaluated the safety and tolerability of OPC-14117, a lipophilic antioxidant, in individuals with HIV-associated cognitive impairment (Table 1) [134]. This study found that OPC-14117 was well tolerated with no significant adverse effects compared to placebo. While there were trends toward cognitive improvement in the treatment group, these changes were not statistically significant, suggesting the need for larger efficacy trials. In a multicenter, randomized, placebo-controlled phase II trial, Schifitto et al. evaluated the safety, tolerability, and efficacy of the selegiline transdermal system (STS), a skin patch formulation of the monoamine oxidase B (MAO-B) inhibitor selegiline, in individuals with HIV-associated cognitive impairment [135]. The primary outcome was the change in neuropsychological performance, measured by the NPZ-6 composite score. The study found that while STS was safe and well tolerated, there were no significant improvements in cognitive performance compared to placebo. These findings suggest that, although STS has a favorable safety profile, its efficacy in improving cognitive function in this population was not demonstrated within the study period. Overall, the findings of these clinical trials are consistent with mechanistic studies suggesting that reducing oxidative stress in vivo may help alleviate neurocognitive symptoms associated with HIV infection. However, antioxidant therapy remains an unproven strategy for treating HAND, and further empirical studies are needed to bridge the gap between in vitro mechanistic findings and in vivo clinical outcomes. Addressing this research question could help establish effective treatments for HAND and deepen our understanding of it. While early antioxidant agents have shown limited clinical efficacy, newer compounds, such as dimethyl fumarate (DMF) that activate endogenous antioxidant defenses via the Nrf2/ARE pathway, demonstrate promising potential [136,137]. However, direct evidence supporting the use of DMF specifically for the treatment of HAND remains lacking. Further research is needed to develop comprehensive, integrative strategies that combine pharmacological therapies with lifestyle interventions to effectively mitigate PLWH.

### 5.2. Poly (ADP-Ribose) Polymerase Inhibitors

Equally important to treating HAND is the urgent need to develop an effective HIV cure strategy. One of the major barriers to achieving a cure is the persistence of latent HIV infection [138,139,140,141,142,143,144]. Notably, microglial cells have been identified as a primary reservoir of HIV within the brain [29,30,31,32,145,146,147]. Although HDAC inhibitors have been extensively studied as latency-reversing agents (LRAs) in the ‘kick-and-kill’ strategy to eliminate HIV reservoir cells, clinical trials have not demonstrated a significant reduction in the size of latent reservoirs [148,149,150]. We recently reported that a combination of FDA-approved cancer drugs targeting poly (ADP-ribose) polymerase (PARP) enzymes with HDAC inhibitors effectively reactivates latent HIV infected cells (Figure 5) [151]. This reactivation enables subsequent elimination of infected cells through the stimulation of immune responses, particularly by natural killer (NK) cells, via a PARP inhibitor. In addition to their role in reducing the latent HIV reservoir, PARP inhibitors that target the PARP pathway have also been reported to suppress neuroinflammation [152]. The PARP protein family consists of 17 members that regulate a wide range of cellular processes, among which PARP-1 is one of the most well-studied and characterized [153,154,155,156]. PARP-1 is a nuclear enzyme that plays a central role in DNA repair and regulates transcriptional responses to genotoxic and oxidative stress through its poly (ADP-ribosyl)ation activity [157,158]. HIV-1 proteins such as gp120 and Tat have been shown to induce oxidative DNA damage indirectly through the generation of ROS, whereas Vpr contributes to DNA damage both by directly causing double-strand breaks and by promoting ROS production [63,105,159,160]. This DNA damage can lead to the activation of PARP-1, which depletes intracellular NAD^+^ and ATP, ultimately leading to neuronal death through a programmed cell death pathway known as parthanatos [161,162,163,164]. Concurrently, PARP-1 activation promotes the expression of proinflammatory genes through the NF-κB pathway, further intensifying neuroinflammation [162,165,166,167]. PARP inhibitors have been shown in vitro to reduce neurotoxicity and inflammatory injury by attenuating ROS production and microglial activation (Figure 5) [168,169,170]. This suggests a potential therapeutic strategy for limiting both direct neuronal damage and secondary inflammation in neurodegenerative and neuroinflammatory conditions. Recent preclinical studies using the SIV-infected macaque model have shown that while PARP-1 expression in the frontal cortex remained unchanged, transcripts of four other PARP family members, PARP9, PARP10, PARP12, and PARP14, were significantly upregulated [171]. In addition, neuronal mitochondria-specific PARP inhibitors could result in a greater neuroprotective effect in the traumatic brain injury (TBI) model mouse [172]. These findings suggest that PARP inhibition may offer therapeutic benefits for neuropathic conditions, supporting interest in their potential application for treating HAND. However, clinical evidence remains limited, and further investigation is required [173]. In broader models of neurodegenerative disease, PARP inhibitors have demonstrated potential in reducing glial activation, mitochondrial dysfunction, ROS production, neuroinflammation, and cognitive decline, suggesting their therapeutic utility may extend beyond HAND. Despite these promising findings, several challenges must be addressed before clinical translation. While several PARP inhibitors have been optimized for high CNS penetration, a critical feature for the effective treatment of brain disorders [174], most clinically approved PARP inhibitors, such as Talazoparib, Niraparib, and Rucaparib, show poor BBB penetration, limiting their use in treating CNS disorders [175]. Additionally, prolonged PARP inhibition raises concerns about genomic stability, particularly in aging individuals living with HIV who may require such treatment. However, we demonstrated that, when combined with HDAC inhibitors, PARP inhibitors can contribute to the reduction in the latent HIV reservoir, suggesting their dual potential in HIV cure strategies as well as in the prevention or mitigation of HAND [151]. Notably, PARP inhibitors are administered only briefly during the reactivation phase of latent reservoir cells to facilitate their elimination, which may minimize the risks associated with prolonged PARP inhibition in PLWH. Although four PARP inhibitors have been approved by the FDA for cancer treatment, further research, particularly pharmacokinetic optimization and clinical trials focused on HIV, is essential to assess the safety and therapeutic efficacy of PARP inhibition.

## 6. Conclusions and Future Directions

Despite the success of cART in controlling systemic HIV replication, HAND persists as a significant burden for PLWH, particularly as the population ages [6,7,8]. From the aforementioned population, approximately 50% of individuals infected with HIV develop some form of neurocognitive disorder [176]. Multiple studies have provided evidence that highlights oxidative stress as a central mechanism in HIV neuropathogenesis. Viral proteins, including gp120, Tat, Nef, and Vpr, in addition to viral RT, induce excessive production of ROS, triggering chronic neuroinflammation, neuronal damage, and ultimately, neurodegeneration. While numerous in vitro studies have provided valuable mechanistic insights into how specific HIV-1 proteins contribute to oxidative stress and neurotoxicity, the physiological relevance of these findings requires careful interpretation. The concentrations of viral proteins used in vitro often exceed those typically present in vivo, particularly in the brains of individuals on suppressive cART, where viral protein levels are expected to be very low [177]. Additionally, many of these studies rely on immortalized cell lines, which may not accurately represent the complex behavior and responses of primary CNS cells, thus limiting the translational relevance of their findings. Nevertheless, in vitro models remain essential tools for dissecting the molecular pathways underlying ROS-mediated damage, offering important context for understanding HAND pathogenesis.

Recent studies have highlighted a growing interest in exploring the link between neuroinflammation and intravenous drug use, particularly given the high prevalence of substance abuse among PLWH. Chronic use of opioids, cannabinoids, and methamphetamines has broadly been shown to induce neurotoxic effects similar to those observed in HIV infection, leading to increased oxidative stress and neuroinflammation [64,178,179,180,181,182]. For instance, research indicates that the inhibition of microglial activation can reduce both neuroinflammatory responses and drug-seeking behaviors [183], suggesting a potential therapeutic target for mitigating substance abuse-related neurotoxicity. Moreover, the co-occurrence of HIV and intravenous drug use is well-documented, with studies reporting significant comorbidity rates, underscoring the need for integrated treatment approaches [184,185]. Substance abuse of the aforementioned drugs may exacerbate the overproduction of ROS, further triggering neuroinflammatory pathways [186,187]. Additionally, certain treatment regimens can necessitate opioid prescriptions for pain management, such as those leading to neuropathic pain, which bears the risk of exacerbating neuroinflammation [188]. Previous research has shown opioid use to be associated with the activation of glial cells and subsequent neuroinflammation, potentially contributing to the development of HAND [188]. Furthermore, recreational drug use is known to compromise BBB integrity by downregulating tight junction proteins, thereby facilitating increased ROS production and neuroinflammation [189,190]. While it is challenging to generalize the effects of opioids, cannabinoids, and methamphetamines on neuroinflammation in PLWH due to their distinct mechanisms of action and metabolites, which can result in variable and sometimes counterintuitive outcomes, substance use remains an important clinical factor. Notably, an estimated 30% of PLWH with HAND have a history of substance use, highlighting the need for substance-specific investigations rather than broad categorizations when evaluating their contribution to neuroinflammation and oxidative stress [187].

Although early antioxidant strategies showed limited clinical efficacy [130,133,134,135], recent advances point to the therapeutic promise of newer compounds that activate endogenous defense pathways, such as dimethyl fumarate via Nrf2/ARE signaling [136,137]. In parallel, PARP inhibitors have emerged as a dual-function strategy, not only enhancing the latency-reversing efficacy of HDAC inhibitors in targeting latent HIV reservoirs as part of potential cure strategies [151], while also showing potential to mitigate neuroinflammation and oxidative damage associated with HAND [152,168,169,170,172,174]. Ultimately, managing HAND requires a multifaceted approach that goes beyond viral suppression. Future therapies will likely involve a combination of antiretroviral treatments, targeted antioxidants, inflammation modulators, and agents addressing HIV latency. Continued research, through well-designed in vivo studies and clinical trials, is essential to translate these mechanistic insights into effective interventions that preserve cognitive function and improve the quality of life for PLWH.

We are currently conducting research to address several of the outstanding questions described above. Ongoing studies are examining ROS expression in microglial cell lines treated with PARP inhibitors, which are also being explored for their ability to activate latent HIV reservoirs. Additionally, there is growing interest in utilizing human microglial cell lines and primary microglial models derived from monocytes, which more accurately reflect the characteristics of CNS-resident microglia in vitro. These models are being used to examine the effects of PARP inhibitors on both latent reservoir reduction and neuroprotection. Further investigation is planned using in vivo models such as EcoHIV-infected mice, a genetically modified HIV construct capable of establishing infection in murine hosts, to study the latency reservoir and HAND [191,192,193,194,195,196,197].

## Figures and Tables

**Figure 1 ijms-26-06724-f001:**
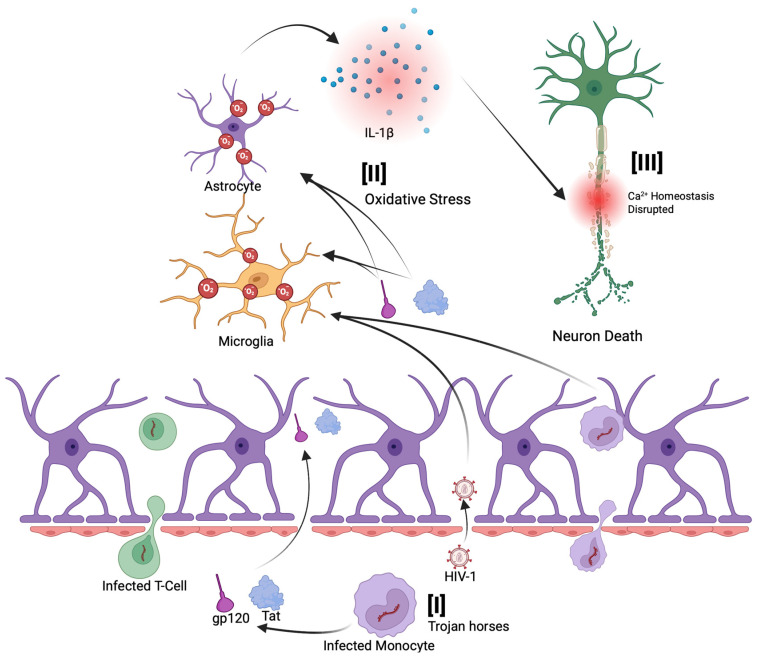
Basic mechanisms of HIV-related neurodegeneration. HIV-associated neurodegeneration begins with viral entry into the central nervous system (CNS) across the blood–brain barrier (BBB). [**I**] Secreted viral proteins increase BBB permeability, allowing HIV-infected monocytes and T cells to infiltrate the CNS, a mechanism often referred to as the “Trojan horse” model. [**II**] Once inside, the infiltrating virus, viral proteins, and infected cells induce oxidative stress in glial cells, leading to the production of proinflammatory cytokines such as interleukin-1β (IL-1β). [**III**] These inflammatory signals disrupt calcium homeostasis in neurons, ultimately contributing to neuronal injury and death.

**Figure 2 ijms-26-06724-f002:**
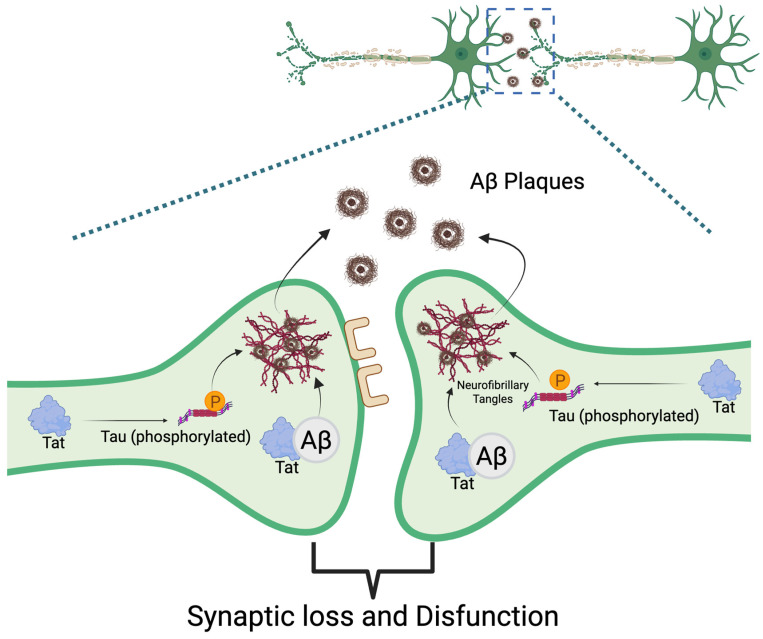
CNS disruption by Tat protein and amyloid-β interaction. Within the CNS, HIV-1 Tat protein binds to amyloid-β (Aβ), disrupting its clearance and promoting its accumulation. Tat also induces hyperphosphorylation of tau proteins, accelerating the formation of neurofibrillary tangles. Together, these effects enhance the development of both intracellular tangles and extracellular Aβ plaques, ultimately leading to synaptic loss and neuronal dysfunction in the CNS.

**Figure 3 ijms-26-06724-f003:**
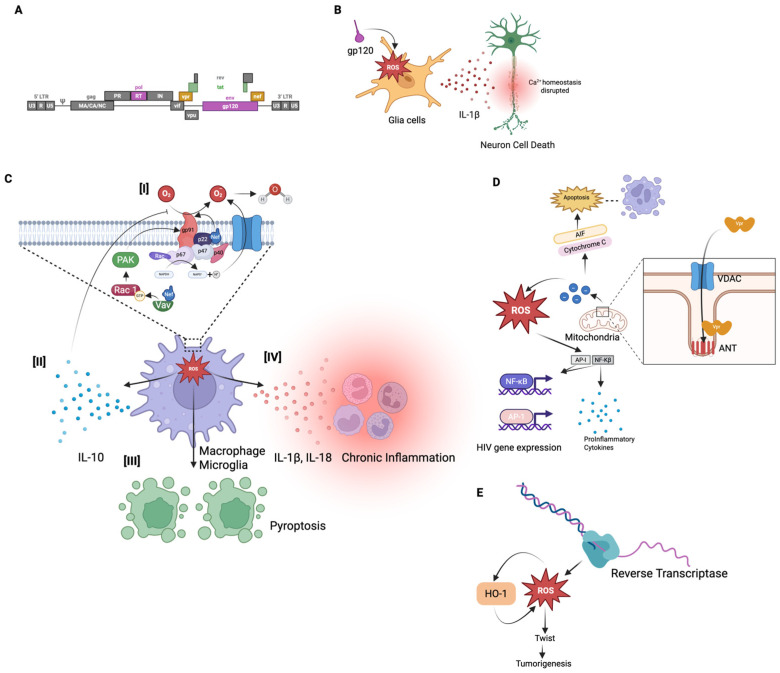
HIV-induced production of reactive oxygen species (ROS). (**A**) Schematic representation of the HIV-1 genome, highlighting viral proteins implicated in ROS generation: gp120, Tat, Nef, Vpr, and reverse transcriptase (RT). (**B**) The envelope glycoprotein gp120 induces intracellular oxidative stress in the glial cells. Elevated ROS in glial cells promotes the release of the proinflammatory cytokine IL-1β, which disrupts calcium homeostasis in adjacent neurons, leading to neuronal injury and death. (**C-I**) The HIV accessory protein Nef interacts with the NADPH oxidase complex via the p22-phox subunit, increasing superoxide anion production and priming the complex for a respiratory burst. Nef also activates the Vav/Rac1/p21-activated kinase (PAK) signaling pathway, further enhancing NADPH oxidase activity. (**C-II**) Nef modulates ROS production in a biphasic manner, ultimately suppressing oxidase activity via IL-10 secretion. (**C-III**) The oxidative environment driven by Nef contributes to inflammasome activation, leading to pyroptosis of uninfected bystander CD4^+^ T cells. (**C-IV**) Elevated superoxide production increases ROS levels, promoting chronic inflammation through cytokines such as IL-1β and IL-18. (**D**) The HIV accessory protein Vpr localizes to the mitochondrial inner membrane, reaching the adenine nucleotide translocator (ANT) through the voltage-dependent anion channel (VDAC). This interaction induces mitochondrial membrane depolarization and permeabilization, resulting in elevated ROS production. The increased ROS promotes the release of pro-apoptotic factors such as cytochrome c and apoptosis-inducing factor (AIF), ultimately leading to apoptosis. Vpr-induced ROS also activates redox-sensitive transcription factors, including NF-κB and AP-1, which enhance HIV gene expression. NF-κB further drives the production of proinflammatory cytokines, contributing to chronic immune activation in PLWH. (**E**) HIV reverse transcriptase (RT), which converts viral RNA into DNA, also contributes to oxidative stress. Its expression in host cells induces ROS production, which is associated with upregulation of the epithelial–mesenchymal transition (EMT) marker Twist. This may promote tumorigenic processes, increased cell motility, and lipid peroxidation. In response to oxidative stress, RT-induced ROS also triggers a cellular antioxidant defense mechanism, including upregulation of heme oxygenase-1 (HO-1), a key detoxification enzyme.

**Figure 4 ijms-26-06724-f004:**
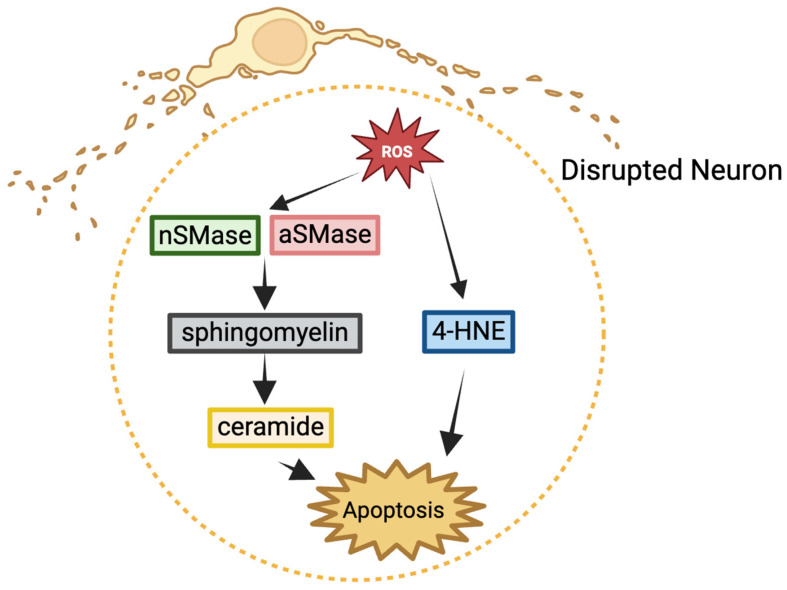
ROS-mediated chronic HIV-associated neurodegeneration. In neurons, ROS triggers increased levels of 4-hydroxynonenal (4-HNE), a marker of lipid peroxidation. ROS also activates acid sphingomyelinase (aSMase) and neutral sphingomyelinase (nSMase), catalyzing the breakdown of sphingomyelin into ceramide. Elevated 4-HNE and ceramide levels promote apoptosis.

**Figure 5 ijms-26-06724-f005:**
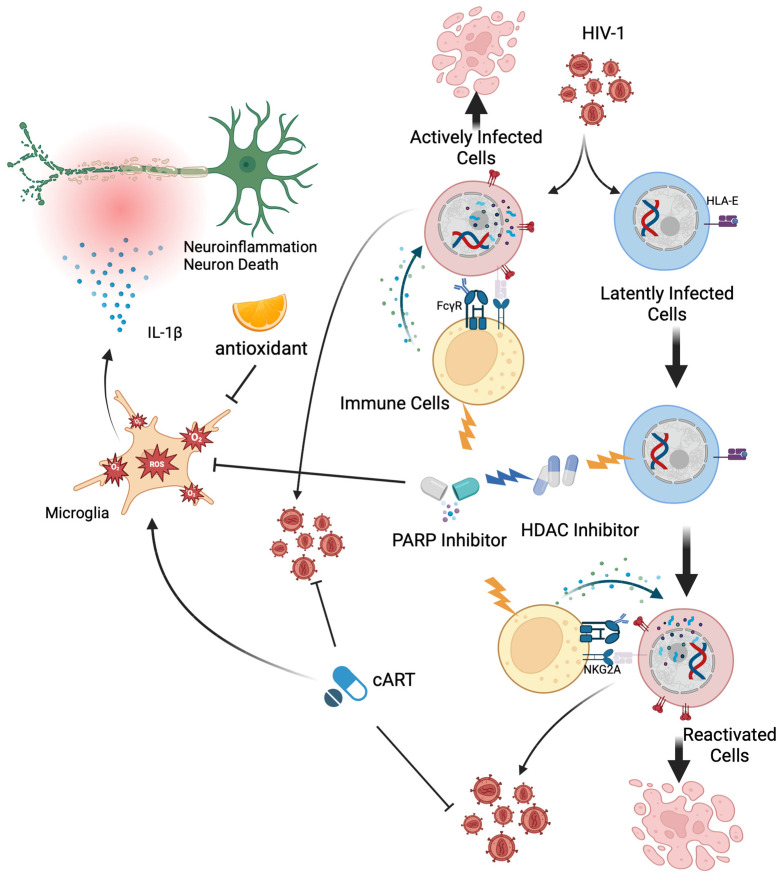
Antioxidants and PARP inhibitors suppress neuroinflammation. Antioxidants and PARP inhibitors mitigate oxidative stress by reducing the production of reactive oxygen species (ROS) in microglia and other CNS cells. This ROS suppression helps alleviate neuroinflammation and prevents subsequent neuronal death. PARP inhibitors also enhance the efficacy of HDAC inhibitor–mediated latency reversal and promote immune activation, particularly of natural killer (NK) cells, to eliminate reactivated HIV-infected cells. Combination antiretroviral therapy (cART) prevents further rounds of viral replication from reactivated cells. However, certain cART components, such as azidothymidine (AZT) and indinavir (IDV), have been shown to induce oxidative stress and mitochondrial dysfunction, underscoring the need for adjunctive strategies to protect CNS integrity.

**Table 1 ijms-26-06724-t001:** Therapeutic Intervention for HIV-Associated Neurocognitive Disorder.

Drug	Mechanism	Status	Reference
NACA	Reduced neuronal death, preserved mitochondrial membrane potential, inhibited oxidative damage	Preclinical in vitro study	[130,132]
Vitamin E and Vitamin C	Reduced oxidative stress via decreased lipid peroxidation (measured by breath pentane, plasma lipid peroxides, and malondialdehyde)	Randomized, double-blind, placebo-controlled trial	[133]
OPC-14117	Scavenges superoxide anion radicals; hypothesized to reduce oxidative neurotoxicity from HIV-infected macrophage-neuron interactions	Phase II—completed and discontinued	[134]
STS	Inhibits monoamine oxidase B (MAO-B); reduces oxidative stress and may promote neurotrophic activity	FDA-approved for major depressive disorder, but not for HAND	[135]
DMF	Activates Nrf2 pathway, reduces oxidative stress, and suppresses microglial-mediated neuroinflammation	No clinical trials in HAND yet; preclinical evidence only	[136,137]

## Data Availability

All data should be referred to in each reference.

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
