# Peer review of "Oxidative Stress in HIV-Associated Neurodegeneration: Mechanisms of Pathogenesis and Therapeutic Targets"

_ijms, 2025, doi:10.3390/ijms26146724_

Round 1
Reviewer 1 Report
Comments and Suggestions for Authors
The manuscript entitled Oxidative Stress in HIV-Associated Neurodegeneration: Mechanisms of Pathogenesis and Therapeutic Targets is a timely and novel review about the role of HIV related ROS in neural inflammation diseases. It streamlined the logic from HIV inducing oxidative stress to ROS induced neuron inflammation and degeneration. Then authors discussed the therapeutic applications of interrupting ROS related pathways. The review cited a wide range of important references, and identified the critical gaps in knowledge, provided important information to the researchers in HIV neuroinflammation field. However, there are still a few points that could be improved before publication. I recommend a minor revision.
- The manuscript is not edited correctly. There are 2 section 3. And repeated sections discussed the same thing. But the 2nd section 3 is much improved. In this report, I will only raise questions for the 2nd section 3 and neglect everything in the first section 3.
- The section 3 is “Mechanisms of ROS production by HIV”. But the authors are mixing mechanisms and evidence. For example, the authors cited reference 78 to prove Env induce ROS when delivered to microglia. But the reference did not discuss why Env induce ROS. It just reported the result. Env is not a transcription factor, so it does not directly affect host gene transcription. And there is no report of Env directly or indirectly binds to ROS related enzymes. So, the mechanism of Env induced ROS is still not clear. IL1b release and neural inflammation could both result from ROS production. Dopaminergic neurons undergo apoptosis when exposed to gp120. All these are evidence of Env induced ROS. They are all downstream of ROS.
On the other hand, the authors did a good job discussing the mechanism of Vpr induced ROS. Vpr interacts with VDAC, leading to mitochondria leakage, which then induces ROS. This is a mechanism.
Mechanisms are important for therapeutic interruptions. If there is a known mechanism, HIV specific drugs could be developed to disrupt specific interactions or activities. Otherwise, one could only use drugs to inhibit general ROS activities, which has way more confounding factors.
The authors could do a better job distinguish the literature, pointing out which papers are real mechanisms of HIV proteins leads to ROS generation, and which papers are evidence of HIV proteins induce ROS in various systems. Evidence is important. It gives people the reason to look for mechanism. But it is not mechanism.
Separating real mechanism and evidence also help the authors get literatures not related to specific HIV proteins out of the section, like reference 86-90. - ROS could result from HIV proteins. It could also result from other factors of HIV infection. For example, the gut-microbiome axis will also affect ROS level in CNS. HIV infection generates viral cDNA, which activates cGAS-STING pathway and generate innate immunity related oxidative stress. ART treatment could also lead to ROS, like protease inhibitors. The authors could include these aspects.
- The authors could make a table summarizing the current clinical trials related to ROS and HIV patients or neural degeneration patients. Now speculative drug effects are mixed with the discussion of drugs under clinical trial. A table could help clarify the content.
- Reference 11 is not relevant.
- Line 126: typo “iscussed”.
Author Response
Reviewer's Comment 1: The manuscript entitled Oxidative Stress in HIV-Associated Neurodegeneration: Mechanisms of Pathogenesis and Therapeutic Targets is a timely and novel review about the role of HIV related ROS in neural inflammation diseases. It streamlined the logic from HIV inducing oxidative stress to ROS induced neuron inflammation and degeneration. Then authors discussed the therapeutic applications of interrupting ROS related pathways. The review cited a wide range of important references, and identified the critical gaps in knowledge, provided important information to the researchers in HIV neuroinflammation field. However, there are still a few points that could be improved before publication. I recommend a minor revision.
Response to the reviewer's comment 1: Thank you for your thoughtful comments and feedback. Our responses to each comment are provided below.
Reviewer's Comment 2: The manuscript is not edited correctly. There are 2 section 3. And repeated sections discussed the same thing. But the 2nd section 3 is much improved. In this report, I will only raise questions for the 2nd section 3 and neglect everything in the first section 3.
Response to the reviewer's comment 2: Thank you for pointing out the duplication of Section 3 in the original manuscript. We have removed the duplicated paragraph 3 and restructured this section from page 6.
Reviewer's Comment 3: The section 3 is “Mechanisms of ROS production by HIV”. But the authors are mixing mechanisms and evidence. For example, the authors cited reference 78 to prove Env induce ROS when delivered to microglia. But the reference did not discuss why Env induce ROS. It just reported the result. Env is not a transcription factor, so it does not directly affect host gene transcription. And there is no report of Env directly or indirectly binds to ROS related enzymes. So, the mechanism of Env induced ROS is still not clear.
IL1b release and neural inflammation could both result from ROS production. Dopaminergic neurons undergo apoptosis when exposed to gp120. All these are evidence of Env-induced ROS.
They are all downstream of ROS.
Response to the reviewer's comment 3: We revised the text in section 3.2, "Envelope Glycoprotein (gp120) and Tat Mediated ROS Production," accordingly.
In addition, we divided the original paragraph 3 into paragraph 3: Mechanisms of Reactive Oxygen Species (ROS) Production by Human Immunodeficiency Virus and paragraph 4: Reactive Oxygen Species (ROS) and Chronic HIV-Associated Neurodegeneration.
Accordingly, Figure 3 has been revised. Figure 3C is now an independent Figure 4, and Figures 3D, 3E, and 3F have been renumbered as Figures 3C, 3D, and 3E, respectively. The revised Figures 3 and 4 are shown on pages 5 and 10, respectively.
Reviewer's Comment 4: On the other hand, the authors did a good job discussing the mechanism of Vpr induced ROS. Vpr interacts with VDAC, leading to mitochondria leakage, which then induces ROS. This is a mechanism
Response to Reviewer's Comment 4: Thank you so much for your positive comment.
Reviewer's Comment 5: Mechanisms are important for therapeutic interruptions. If there is a known mechanism, HIV specific drugs could be developed to disrupt specific interactions or activities. Otherwise, one could only use drugs to inhibit general ROS activities, which has way more confounding factors.
The authors could do a better job distinguish the literature, pointing out which papers are real mechanisms of HIV proteins leads to ROS generation, and which papers are evidence of HIV proteins induce ROS in various systems. Evidence is important. It gives people the reason to look for mechanism. But it is not mechanism.
Separating real mechanism and evidence also help the authors get literatures not related to specific HIV proteins out of the section, like reference 86-90.
Response to Reviewer's Comment 5: According to the reviewer's comment, we revised subsection 3.2, "Envelope Glycoprotein (gp120) and Tat Mediated ROS Production," to clarify the observed mechanisms of ROS induction by gp120. In addition, we further explored the mechanistic details of gp120-mediated ROS production from lines 228 to 247 on page 7.
Reviewer's Comment 6: ROS could result from HIV proteins. It could also result from other factors of HIV infection. For example, the gut-microbiome axis will also affect ROS level in CNS. HIV infection generates viral cDNA, which activates cGAS-STING pathway and generate innate immunity related oxidative stress. ART treatment could also lead to ROS, like protease inhibitors. The authors could include these aspects.
Response to Reviewer's Comment 6: According to the reviewer's comment, we additionally discuss the HIV protein-independent induction of ROS production and oxidative stress on page 6, lines 169-180.
Reviewer's Comment 7: The authors could make a table summarizing the current clinical trials related to ROS and HIV patients or neural degeneration patients. Now speculative drug effects are mixed with the discussion of drugs under clinical trial. A table could help clarify the content.
Response to Reviewer's Comment 7: We created and integrated Table 1 into the manuscript according to the reviewer's suggestion.
Reviewer's Comment 8: Reference 11 is not relevant.
Response to Reviewer's Comment 8: We replaced reference 11 with a relevant article.
Reviewer's Comment 9: Line 126: typo “iscussed”.
Response to Reviewer's Comment 9: We fixed the typo.
Reviewer 2 Report
Comments and Suggestions for Authors
Overall, this is systematic insightful and informative review highlighting the potential role of the ROS and therapies to counter oxidative inflammation. A good background is supplied, mechanisms are shown in figures, and studies are well described with respect to In Vivo versus Ex Vivo and cell type or cell line employed. However, it is also this comprehensive analysis that raises issues for those of us who have been in the HIV field for some time.
1. One concern involves the effect of specific viral proteins (gp 120 and TAT) on ROS and inflammation. From a background on viral protein detection and working closely with those performing In Vitro studies, we know that levels of gp120 In Vivo (especially in the brain) are extremely low. Ex Vivo effects are based on high input levels, have been inconsistent from study to study, and vary according to source of gp120 and even batch. I would suggest that the authors preface the section by mentioning the difficulty in extrapolating exposure doses from In Vivo to In Vitro. Few in the field would question that HIV proteins are expressed in the CNS in those on cART, but the effects could be more general (foreign or missfolded protein effects) rather than mediated by gp120 (or VPR or NEF) toxicity. Specifically, it can be noted that In Vitro studies seldom include an unrelated HIV or other protein control and no studies are reported for HIV capsid proteins that are produced in greatest quantities.
2. Similarly, the data regarding TAT are problematic in that, according to the latency literature, TAT is a marker of active viral replication and is relatively labile. From that perspective, the HIV latency community doubts the TAT expression reported in the NeuroHIV community. Within lab assays for TAT have not been evaluated by outside sources. In our experience, TAT levels measured by our colleagues did not correlate with HIV RNA (TAT was (+) when HIV RNA by PCR was undetectable), leading us not to include collaborative TAT data in our manuscripts.
3. A related issue is potential misinformation resulting from use of transformed T-cell, monocytoid and other cell lines for HIV studies, which led to major controversies and failure of large clinical trials in the early years of HIV research. The fact that abstracts for CROI discourage studies with cell lines reflects the importance of this issue. A review cannot and should not ignore all such published reports, but should also include limitations, cautioning against over interpretation of studies based on cell lines and noting that there is justified mistrust in the scientific community.
4.In terms of the “kick and kill” approach to HIV cure, the authors might mention that clinical trials with HDAC inhibitors have failed to show reduction in HIV reservoir and residual latently infected cells appear to be relatively resistant to lysis.
5.In the “conclusions” section, there is discussion of substance abuse increasing inflammation. As part of a group that has examined specific effects of opioids, cannabinoids, and methamphetamine on PLHIV, I would caution against any generalizations since they reveal practical inexperience. Each substance has specific effects on various cell types and metabolites of each substance all complicate the picture, often against general predictions. As an example, many researchers predicted that PLHIV who used methamphetamine would suffer severe COVID during the pandemic based on the assumption that it is broadly “pro-inflammatory”. In fact, none of our users experienced severe COVID, consistent with unexpected data that methamphetamine can reduce monocyte cytokine expression and pseudophedrine (1 methyl group away from methamphetamine) appears to be an effective treatment for COVID.
Author Response
Reviewer's Comment 1: Overall, this is systematic insightful and informative review highlighting the potential role of the ROS and therapies to counter oxidative inflammation. A good background is supplied, mechanisms are shown in figures, and studies are well described with respect to In Vivo versus Ex Vivo and cell type or cell line employed. However, it is also this comprehensive analysis that raises issues for those of us who have been in the HIV field for some time.
Response to the reviewer's comment 1: Thank you for your thoughtful comments and feedback. Our responses to each comment are provided below.
Reviewer's Comment 2: One concern involves the effect of specific viral proteins (gp 120 and TAT) on ROS and inflammation. From a background on viral protein detection and working closely with those performing In Vitro studies, we know that levels of gp120 In Vivo (especially in the brain) are extremely low. Ex Vivo effects are based on high input levels, have been inconsistent from study to study, and vary according to source of gp120 and even batch. I would suggest that the authors preface the section by mentioning the difficulty in extrapolating exposure doses from In Vivo to In Vitro. Few in the field would question that HIV proteins are expressed in the CNS in those on cART, but the effects could be more general (foreign or missfolded protein effects) rather than mediated by gp120 (or VPR or NEF) toxicity. Specifically, it can be noted that In Vitro studies seldom include an unrelated HIV or other protein control and no studies are reported for HIV capsid proteins that are produced in greatest quantities. Similarly, the data regarding TAT are problematic in that, according to the latency literature, TAT is a marker of active viral replication and is relatively labile. From that perspective, the HIV latency community doubts the TAT expression reported in the NeuroHIV community. Within lab assays for TAT have not been evaluated by outside sources. In our experience, TAT levels measured by our colleagues did not correlate with HIV RNA (TAT was (+) when HIV RNA by PCR was undetectable), leading us not to include collaborative TAT data in our manuscripts.
Response to the reviewer's comment 2: According to the reviewer's comment, we described the discrepancy in viral protein concentration in vivo versus the in vitro assay on page 16, lines 592-597.
Reviewer's Comment 3:A related issue is potential misinformation resulting from use of transformed T-cell, monocytoid and other cell lines for HIV studies, which led to major controversies and failure of large clinical trials in the early years of HIV research. The fact that abstracts for CROI discourage studies with cell lines reflects the importance of this issue. A review cannot and should not ignore all such published reports, but should also include limitations, cautioning against over interpretation of studies based on cell lines and noting that there is justified mistrust in the scientific community.
Response to the reviewer's comment 3: In addition to the discrepancy in viral protein concentration, we also pointed out the limitation of cell lines in the same section on page 16, lines 597-602.
Reviewer's Comment 4: In terms of the “kick and kill” approach to HIV cure, the authors might mention that clinical trials with HDAC inhibitors have failed to show reduction in HIV reservoir and residual latently infected cells appear to be relatively resistant to lysis.
Response to the reviewer's comment 4: According to the reviewer's suggestion, we described it on page 15, lines 532-535.
Reviewer's Comment 5: In the “conclusions” section, there is discussion of substance abuse increasing inflammation. As part of a group that has examined specific effects of opioids, cannabinoids, and methamphetamine on PLHIV, I would caution against any generalizations since they reveal practical inexperience. Each substance has specific effects on various cell types and metabolites of each substance all complicate the picture, often against general predictions. As an example, many researchers predicted that PLHIV who used methamphetamine would suffer severe COVID during the pandemic based on the assumption that it is broadly “pro-inflammatory”. In fact, none of our users experienced severe COVID, consistent with unexpected data that methamphetamine can reduce monocyte cytokine expression and pseudophedrine (1 methyl group away from methamphetamine) appears to be an effective treatment for COVID.
Response to the reviewer's comment 5: According to the reviewer's suggestion, we described the challenges of generalizing drug effects on neuroinflammation on pages 16-17, lines 621-627.